# Integrating Computer Vision and CAD for Precise Dimension Extraction and 3D Solid Model Regeneration for Enhanced Quality Assurance

Binayak Bhandari [1,2,*] and Prakash Manandhar [3,4,5]

1   School of Mechanical and Manufacturing Engineering, University of New South Wales,
    Sydney, NSW 2052, Australia
2   Department of Railroad Integrated Systems, Woosong University, Daejeon 34606, Republic of Korea
3   Simulate Anything, Inc., Hopkinton, MA 01748, USA; prakashm@alum.mit.edu
4   Wentworth Institute of Technology, Boston, MA 02115, USA
5   Massachusetts Institute of Technology, Cambridge, MA 02139, USA
*   Correspondence: binayak.bhandari@unsw.edu.au or thebinayak@gmail.com

**Abstract:** This paper focuses on the development of an integrated system that can rapidly and accurately extract the geometrical dimensions of a physical object assisted by a robotic hand and generate a 3D model of an object in a popular commercial Computer-Aided Design (CAD) software using computer vision. Two sets of experiments were performed: one with a simple cubical object and the other with a more complex geometry that needed photogrammetry to redraw it in the CAD system. For the accurate positioning of the object, a robotic hand was used. An Internet of Things (IoT) based camera unit was used for capturing the image and wirelessly transmitting it over the network. Computer vision algorithms such as GrabCut, Canny edge detector, and morphological operations were used for extracting border points of the input. The coordinates of the vertices of the solids were then transferred to the Computer-Aided Design (CAD) software via a macro to clean and generate the border curve. Finally, a 3D solid model is generated by linear extrusion based on the curve generated in CATIA. The results showed excellent regeneration of an object. This research makes two significant contributions. Firstly, it introduces an integrated system designed to achieve precise dimension extraction from solid objects. Secondly, it presents a method for regenerating intricate 3D solids with consistent cross-sections. The proposed system holds promise for a wide range of applications, including automatic 3D object reconstruction and quality assurance of 3D-printed objects, addressing potential defects arising from factors such as shrinkage and calibration, all with minimal user intervention.

**Keywords:** vision-based measurement; automation; computer vision; 3D model regeneration; Computer-Aided Design; Internet of Things

## 1. Introduction

In today's technologically driven world, images play a pivotal role in conveying information and enabling machine understanding. Computer vision (CV) is the key to unlocking the information hidden within these images, empowering machines to recognize, analyze, and reconstruct data from visual inputs. The applications of CV are far-reaching, impacting fields as diverse as automation, robotics, healthcare, and quality assurance [1]. Bhandari and Lee [2] showed a functioning robotic hand with vision-based shape analysis and tactile sensing-based object identification for distinguishing soft and hard objects by comparing the contours of the objects.

One of the fundamental challenges in extracting valuable information from images is the necessity of image segmentation: dividing an image into meaningful segments or objects. Image segmentation forms the foundation upon which various image processing

tasks rest, from object recognition to tracking and, notably, the reconstruction of three-dimensional (3D) models from image and video data [3].

Central to the success of these applications is the ability to identify and analyze the shapes and contours of segmented regions. Contour detection, a cornerstone in computer vision, provides crucial information about the object's shape, which, in turn, serves as the basis for the recognition system. The precision and accuracy of contour detection are pivotal in ensuring the retrieval of high-quality information from the visual data. In particular, robots require 3D information about the grasping object in an unstructured industrial environment [4].

The present study is focused on the integration of computer vision and Computer-Aided Design (CAD) to achieve precise dimension extraction and 3D solid model regenerations. These aspects are critical components in the field of quality assurance, enabling the thorough analysis of products and structures in three dimensions. The paper further explores the significance of this approach with case studies addressing existing challenges, thereby contributing to the evolving field of computer vision for enhanced quality assurance.

## 2. State of the Art

Three-dimensional model generation is one of the most significant topics in computer vision and thus has received significant attention in the past few decades. A 3D model serves as a digital representation of a physical object necessary for 3D printing. The two major representations of 3D models are (a) a solid model and (b) a surface model. Solid models define the volume of the object and are widely used in engineering simulations, while surface models represent the contours of an object which are important for defining model appearance, such as automotive design, entertainment industries, and aerospace industries [5].

Converting a solitary 2D image into a 3D model has proved to be a challenging task for developers and has posed a significant research challenge across various fields. Intwala [6] explored various ways of extracting drawing features from 2D CAD drawings' raster images and reported that the algorithm can be used to generate a DXF CAD file with the possibility to be edited by the user. The main objective of this research was to assist in digitizing the legacy hard copy of CAD drawings. Meanwhile, Fan [7] worked on descriptive geometry of 2D CAD drawings, including understanding sectional and assembly drawings.

Three-dimensional reconstruction, such as human face and head reconstruction, has large applications in realistic 3D animation. This requires capturing a large number of images of the face at varying angles, lighting conditions, and facial expressions [8]. Researchers have successfully employed Convolution Neural Networks (CNNs) to train algorithms for generating 3D models from 2D facial images [9].

Integrated Computer-Aided Design (CAD) and additive manufacturing enable the rapid prototyping of 3D models with minimal post-processing and improved mechanical properties. Patil et al. [10] developed some image processing techniques to extract geometric features that optimize the laser additive manufacturing process. However, traditional CAD modeling can be a tedious, time-consuming, and error-prone (observational errors, instrumental errors, and environmental errors) process, often lacking essential dimensions to generate 3D solid models. Oftentimes, the measured dimensions might not be sufficient, and missing important dimensions can potentially make the generation of the 3D models impossible.

Generating high-quality 3D CAD models can also help analyze machinability and manufacturing. Peddireddy et al. [11] reported that being able to generate a triangularly tessellated surface (STL) from the 3D CAD model can be used to predict the manufacturability of the CAD design and machining processes. However, working with highly complex surfaces is a constant challenge for designers. Fan et al. [12] presented a reverse engineering

method based on a CAD model and point cloud model because of the difficulty in dealing with Non-Uniform Rational B-Splines (NURBs) surfaces.

To address the inherent limitations in computer vision for 3D model generation and the time-consuming nature of manual CAD modeling, the integration of a computer vision algorithm and a CAD system has emerged as a promising solution. This integration leverages the strengths of each method to compensate for their respective weaknesses.

The central focus of this research is to devise a cost-effective solution to overcome these limitations by harnessing the power of computer vision algorithms, the Internet of Things (IoT), and a commercial CAD package to seamlessly regenerate 3D solid models within a CAD environment. To make this happen, an integrated system was developed that combines computer vision with CAD capabilities that efficiently extract essential information from images and automates the 3D solid model generation processes within the CAD software CATIA V5.

## 3. Methodology

In the course of these efforts, several intricate tasks were addressed, such as the precise control of objects via robotic manipulation to optimize camera angles, image segmentation, image transformation and realignment, contour detection, virtual measurements, exporting contour coordinates, and the automatic generation of a computer program based on these coordinates. These processes are designed to seamlessly interface with a specialized commercial CAD system to produce comprehensive three-dimensional models.

Two sets of experiments were performed: one with a simple cubical object, and the other with a more complex geometry that needed various measurements to redraw it in the CAD system. The dimensional accuracy of the real and virtual objects was compared, and the surface roughness of the regenerated object was analyzed after 3D printing. The schematic diagram showing processes for the auto-generation of objects is shown in Figure 1. One of the main contributions of the research is the development of an integrated system for regenerating a 3D solid with uniform thickness. Though the developed system has specific applications as it is limited to the regeneration of 3D objects with uniform cross-sections, it has huge potential applications in automatic 3D object reconstruction and in the quality assurance of 3D-printed objects with minimal user intervention.

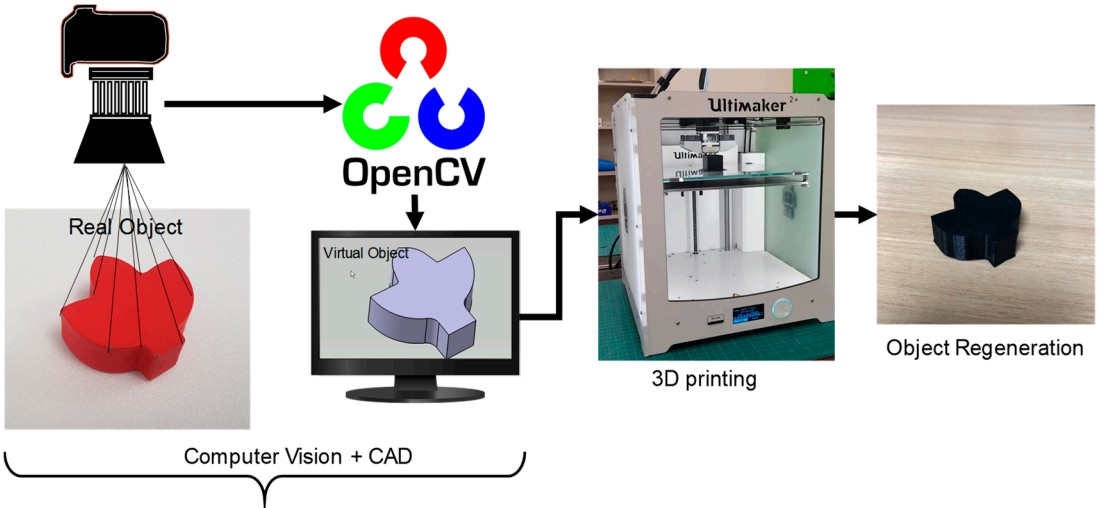

**Figure 1.** Schematic diagram showing the process of transforming real objects to virtual objects which can be used for 3D printing.

## 4. Model Description

### 4.1. Robotic Hand Design

The Hackberry open-source 3D-printable bionic arm was used in this study [13] which has gathered huge attention from developers, designers, and artificial arm users

in recent days. The Hackberry bionic hand is an active prosthesis with an electrically powered hand controlled via Arduino Micro technology. The assembled robotic hand has variable movements suitable for grabbing objects and changing the orientation. The hand was printed by the 3D printer Ultimaker 2+ using an ABS filament with a circuit board and a rechargeable battery embedded in the hand, making it versatile and appealing to researchers.

### 4.2. Control

A small single-board computer developed by the Raspberry Pi Foundation called Raspberry Pi 3 Model B [14] and a MotionEyeOS were used for this study. The MotionEyeOS is a Linux distribution that turns a Raspberry Pi into a video surveillance system [15]. Out of the many features of MotionEyeOS, it is easy to set up, connects to a local network using Ethernet or Wi-Fi, and is compatible with USB cameras. Figure 2 shows the implementation of Raspberry Pi with MotionEyeOS for image acquisition.

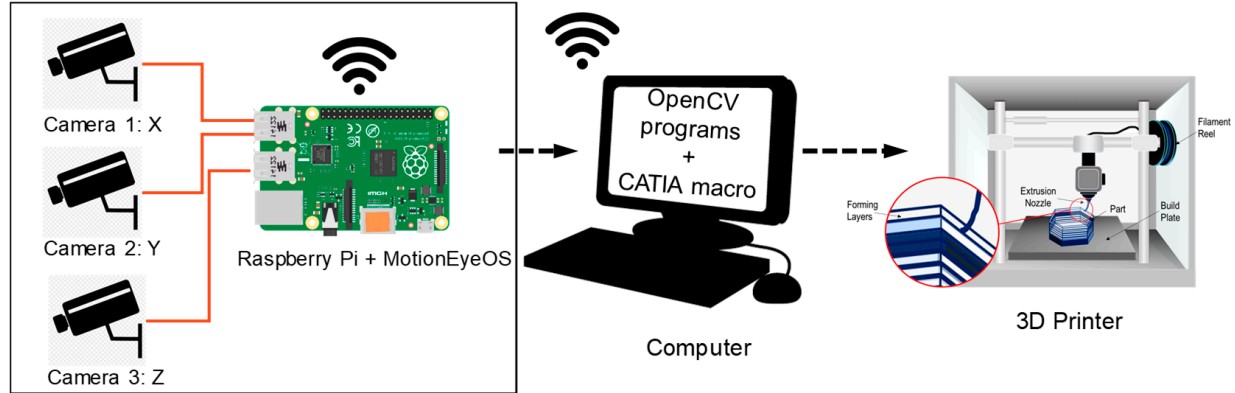

**Figure 2.** Schematic working principle.

### 4.3. Cameras

Three cameras were mounted on three axes (X, Y, Z) to capture the front, top, and side views of the object. General-purpose webcams as cameras were used for capturing images of the object. Logitech c270 model webcams were selected as they are inexpensive and produce high-definition images. However, because of the lack of an autofocus function in the webcams, focus was performed manually. While only two cameras suffice for capturing two views for extracting the surface and thickness data, the system is designed with additional cameras for future advanced research work.

### 4.4. System Integration

The experiment test frame was designed to fix all the hardware in the proper position to conduct the experiment. The robotic hand was used to hold the object at the required position and angle. In addition to fine-tuning the angle and position of the object, it was possible to adjust the gripping force in the robotic hand. The robotic hand position was adjusted by fine-tuning the wrist rotation button. Three webcams were positioned on the three perpendicular axes, X, Y, and Z, as shown in Figure 3. The cameras were interfaced with the Raspberry Pi 3 to take images of the object held by the robotic hand. This helped to capture images of the object in three perpendicular directions simultaneously. This method was efficient and less time-consuming than acquiring images one at a time.

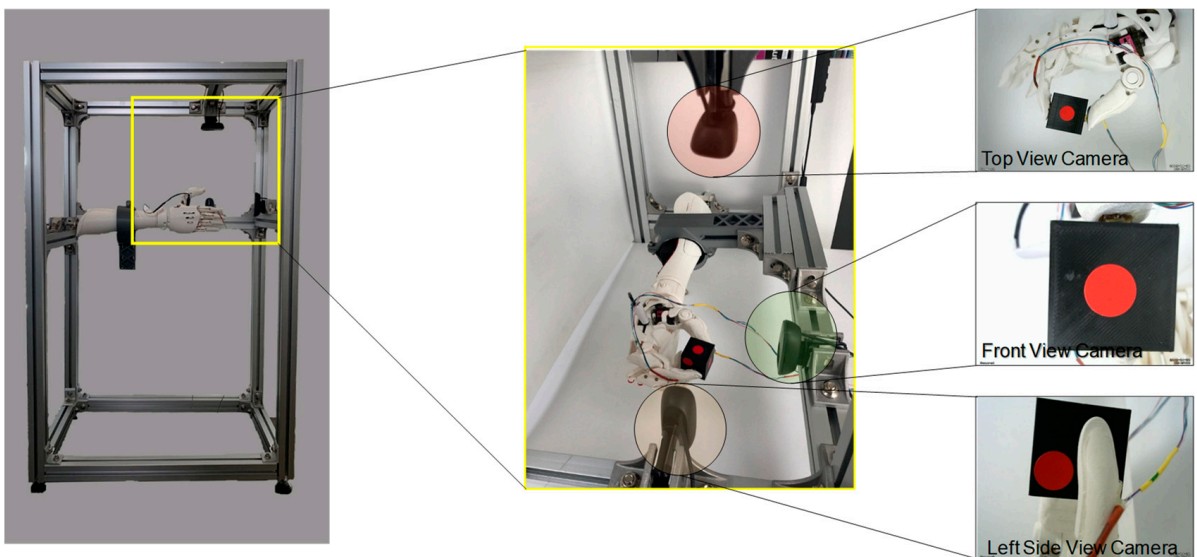

**Figure 3.** Experimental set-up with three cameras and a robotic hand holding the object.

## 5. Methodology of Extracting Coordinate Details

### 5.1. Image Segmentation

The 3D construction uses scanned data and 2D images taken by rotating the object in 360 degrees. Thus, this approach has no background; however, it is time consuming and often expensive. Unlike the above approach, this research focuses on the image taken by an ordinary webcam (or camera) with the background. The problem of separating the object from the background (image segmentation) in a complex environment was performed by using a powerful GrabCut [16,17] segmentation algorithm. This algorithm is popular for efficient and interactive foreground/background segmentation of images where the user is required to draw a rectangle around the object to be cut. The GrabCut is built on the foundation of GraphCut [18] which is a powerful optimization technique to achieve robust image segmentation. The GrabCut uses a Gaussian Mixture Model (GMM) to model foreground and background.

To help with segmentation, a trimap $T = \{T_B, T_U, T_F\}$ is used where $T_B$ (alpha value $\alpha_n = 0$), $T_U$ and $T_F$ (alpha value $\alpha_n = 1$) stores background, unknown, and foreground pixels. GrabCut performs iterations to refine the labeling of the foreground and background pixels. All of the pixels in $T_U$ are assigned a cluster based on the minimum unary weighing function $D(n)$ [19].

$$D(n) = -log\sum_{i=1}^{K} \pi(\alpha_n, i) \frac{1}{\sqrt{det\sum(\alpha_n, i)}} e^{(-\frac{1}{2}[Z_n - \mu(\alpha_n, i)]^T \sum(\alpha_n, i)^{-1}[Z_n - \mu(\alpha_n, i)])} \qquad (1)$$

where $K$ is the number of clusters, $\mu$ is the mean RGB value, $\pi$ is the weighing coefficient, and $Z$ is the RGB row vector of $n$.

Figure 4 shows the process of how the object was segmented from the 2D image: Figure 4a shows the image of the object held by the robotic hand, Figure 4b shows a user-created rectangle that is used to create a trimap for the GrabCut algorithm, and Figure 4c shows the segmented object from the image. The blue box in the second image is a region of interest (ROI) and the black patch is a manually marked mask denoting the background color. Detailed operations and finer touch-ups of the GrabCut algorithm can be found in [11].

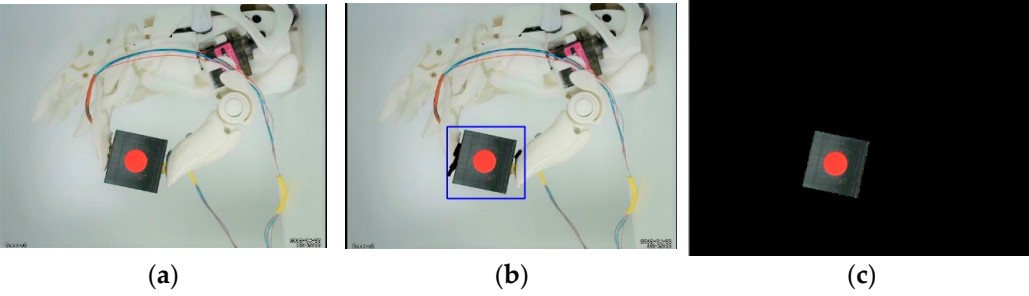

(**a**)  (**b**)  (**c**)

**Figure 4.** The object before and after the Grabcut Algorithm (**a**) An object held by the robotic arm, (**b**) the region of interest marked by the blue rectangle, and (**c**) the segmented image after it was processed by the Grabcut algorithm.

*5.2. Image Transformation*

Segmented images could be inclined and/or skewed as shown in Figure 5 (left). In order to align the segmented image to the perpendicular axis, the four edge points were used to apply the perspective transformation to obtain a top-down "bird eye view" [20] of the image as shown in Figure 5 (right).

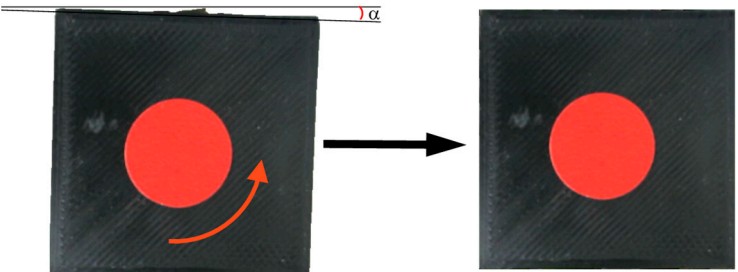

**Figure 5.** The skewed image (**left**) and perspective-transformed image (**right**).

*5.3. Contours and Photogrammetry*

Circular red stickers with a known dimension were used as reference objects in the top and front views. A Canny edge detector [21] was employed to detect edges in the images, and contour gaps were filled using dilation and erosion functions. For hysteresis thresholding in the Canny edge detection algorithm, two threshold values were utilized: minVal and maxVal. Any edges with an intensity gradient greater than maxVal are considered edges, while those below minVal are deemed non-edges and are therefore discarded. Segmented image had at least two contours (a) of the object and (b) the reference shape (red circle). It is clear that the largest contour represents the object, while the smaller contour represents the reference shape. The sample code snippet is provided in Table 1.

**Table 1.** Code snippet of the contour detection in the image.

```python
# Perform edge detection, dilation + erosion to close gaps
edged = cv2.Canny(gray, 50, 100)# image, minVal, maxVal
edged = cv2.dilate(edged, None, iterations=2)
edged = cv2.erode(edged, None, iterations=2)
# Find contours
cnts                                                         =
                        cv2.findContours(edged.copy(),cv2.RETR_TREE,cv2
                        .CHAIN_APPROX_SIMPLE)
```

Figure 6 shows the original object with a marker with known dimension and the complex cross-sectional object.

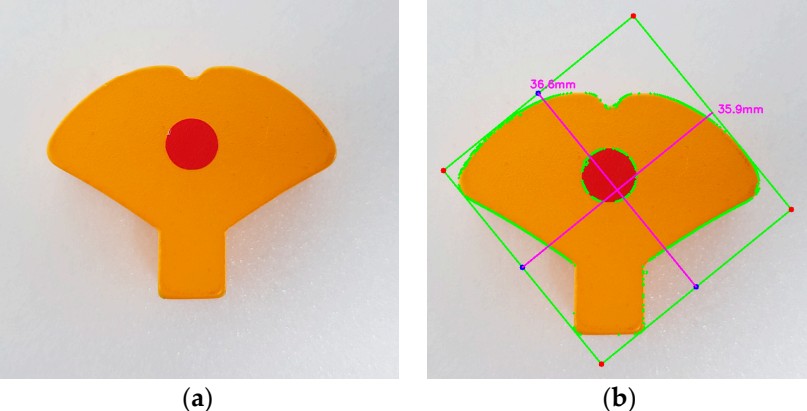

|      |      |
| :--: | :--: |
| (**a**) | (**b**) |

**Figure 6.** (**a**) A complex cross-sectional object with the reference marker and (**b**) a bounding box enclosing the object and the contour points.

## 6. Result and Discussion

The three-dimensional views or pictorial drawings are not able to show all the details of the object such as holes and grooves. In order to obtain more complete information about the object, an orthographic projection (or multi-view drawings) is generally used in engineering design, as shown in Figure 7. Orthographic views are two-dimensional views of three-dimensional objects created by projecting a view of an object onto a plane parallel to one of the planes of the object. Technical drawings usually include the front, top, and side orthographic views. However, two views are sufficient for uniform cross-sectional and cylindrical bodies. In the case of complicated bodies with slots and holes, an additional sectional view might be needed.

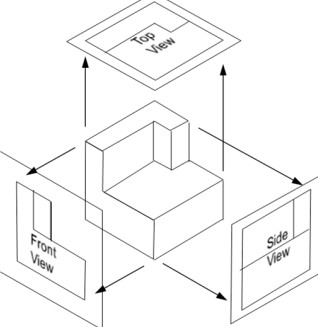

**Figure 7.** Orthographic views of a 3D object showing front, top, and side views.

Integrating all three views within a CAD environment to generate a 3D model without adequate training is not a straightforward task. In this study, we employed a Computer-Aided Three-dimensional Interactive Application (CATIA V5) for 3D model generation. Although CATIA V5 [22] is widely recognized as one of the most popular engineering and design software programs, it still demands the expertise of a skilled engineer to create intricate 3D shapes. To streamline the process of generating a 3D model from contour information (x, y, and z coordinates), we developed a Visual Basic Script macro (CATScript). Figure 8 illustrates the detailed algorithm for integrating previous steps with a CAD system in this context.

The flowchart illustrates the comprehensive system architecture that outlines the process of generating a contour of an object from input images captured by the cameras. The workflow commences on the input of the object's image, followed by meticulous background removal using the GrabCut algorithm. Subsequently, edge detection is applied to the processed image, resulting in the extraction of key object edges. These detected edges are utilized to construct the object's contour.

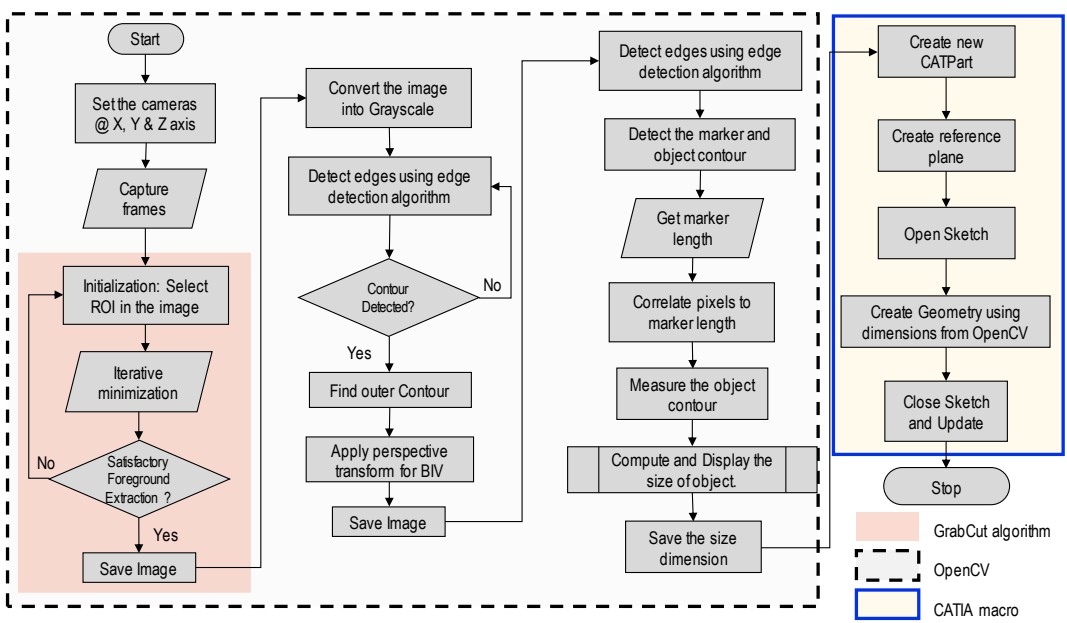

**Figure 8.** An integrated flowchart for extracting object dimensions in OpenCV and creating geometry in CATIA V5.

Moreover, the contour's accuracy is ensured by referencing marker dimensions, facilitating precise measurement of both its x and y coordinates. The coordinate data are then preserved in a *.txt file, concluding the operation using OpenCV. A custom Python program is employed to read the x and y dimensions from the text file, subsequently generating a macro (*CATScript) in VBScript programming language. This generated macro can be directly imported into CAD software, where it aids in sketching a 2D surface that can be extruded to the user's specified dimensions, ultimately producing a 3D solid model.

*Case Studies*

Two sets of experiments were performed to validate the proposed process of 3D model generation using (a) a relatively simple cubical object and (b) a more complicated irregular-shaped cross-section object. Both experiments followed a similar procedure as shown in Figure 9a.

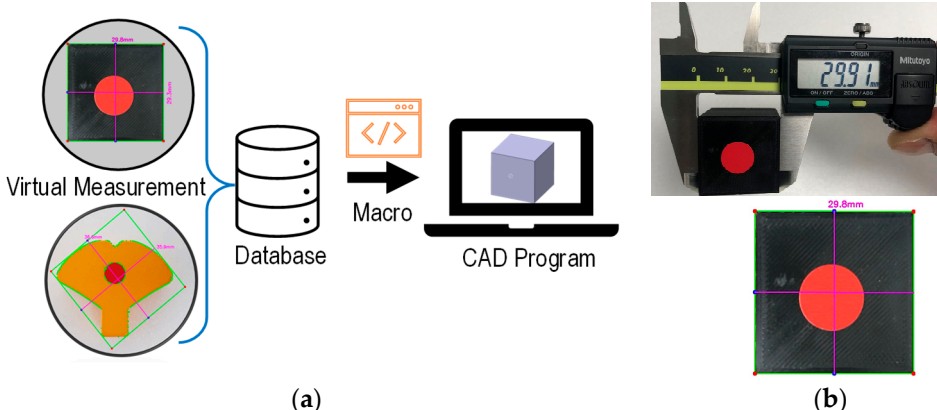

(**a**)                    (**b**)

**Figure 9.** (**a**) Object coordinate extraction to 3D model generation steps and (**b**) physical and virtual measurement comparisons.

In order to measure the size of an object in an image, the program first performs calibration using information about the reference object. For calculating 'pixel/metric', the ratio of the pixel of a known reference object to its known dimensions was used.

$$Pixels\ per\ metric = \frac{object\ width\ in\ pixel}{known\ width}$$

In the case of a simple cubical object, the coordinates of two end points of a rectangular surface suffice for the generation of a rectangular shape; similar processes can be repeated to generate other sides.

The comparison of the real and virtual measurements is shown in Figure 9b. For this specific case, the error (%) is calculated as shown below.

$$Percentage\ error = \frac{29.9 - 29.8}{29.9} \times 100\% = 0.3\%$$

It can be seen that the discrepancy between the two is less than 1%, which is acceptable in many practical scenarios. This discrepancy can further be reduced by taking images from the camera at a perfect 90-degree angle, using higher resolution cameras, sharp-focused images, and the proper selection of ROI for GrabCut Algorithm operation.

The complex-shaped object in the second case produced a large number of continuous points along the boundary of the contour. The contour pixel coordinates $(p(x), p(y))$ were transformed to the $(x, y)$ coordinate using the reference dimension $(\phi)$ and the corresponding pixel width $(\Delta)$ using the following relation:

$$(x, y) = \left( \frac{\phi}{\Delta} \times p(x) \right), \left( \frac{\phi}{\Delta} \times p(y) \right)$$

The dimensions of the object were exported to Computer-Aided Design (CAD) software via a macro [23] to generate a 3D model. A separate Python program was developed to automatically generate a segment of the macro program (*.CATScript) using the coordinate data. In the final step, the macro was executed within the CAD program to produce a sketch. This sketch had a few open contours requiring minor geometry adjustments. The resulting CAD model displayed an almost imperceptible level of roughness in its contour profile, measuring in the order of micrometers (~μm).

Figure 10a presents a plot of points within the CAD environment, which, upon extrusion, resulted in the creation of a 3D solid model depicted in Figure 10b.

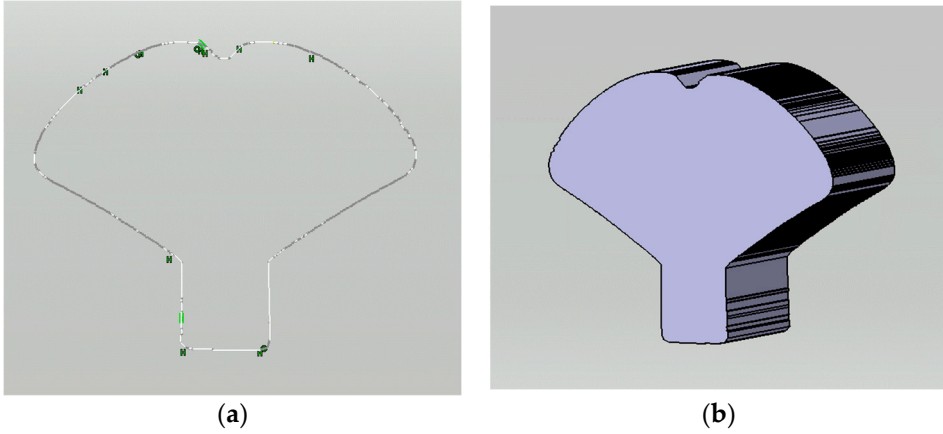

(**a**)  (**b**)

**Figure 10.** Comparison of a real object and steps of the CAD generation process. (**a**) The sketch of the contour point in the CAD environment; (**b**) isometric view of the object generated by the CAD program.

The generated CAD file is saved on the disk as a stereolithography CAD file (*.stl), which is imported into the open-source slicing software Ultimaker Cura for 3D printing, as shown in Figure 11a. This slicing software helps to set various printing parameters, such as model position and orientation, infill density, layer deposition thickness, and the selection of materials, and gives information on the estimated time of realization, and the weight and length of the filament. Finally, Figure 11b shows the direct comparison between the original object (in yellow) and the 3D-printed counterpart (in black), with the two overlaid for visual analysis.

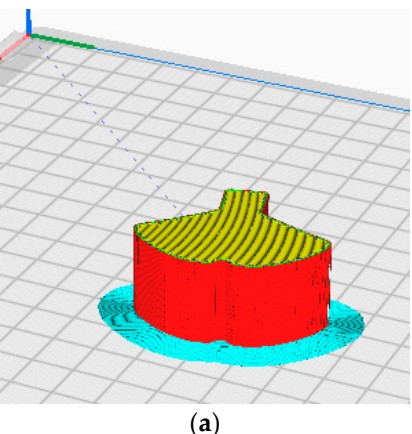
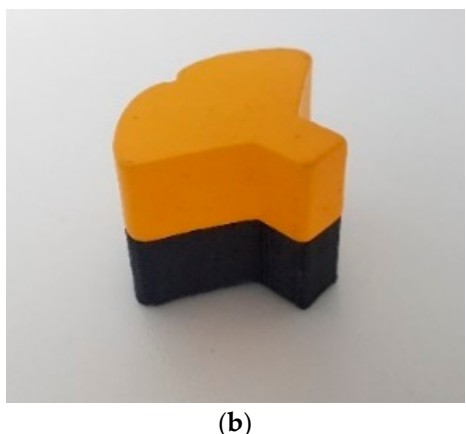

| (**a**) | (**b**) |

**Figure 11.** (**a**) G-Code file generation in Ultimaker Cura (cyan is support, red is the object shell and yellow shows the layer infill) and (**b**) Comparison of the 3D printed object with the original object.

## 7. Conclusions

This research successfully accomplished the generation of a 3D model from images within a commercial CAD program, demonstrating its efficacy for both simple cubic shapes and complex objects. This achievement was made possible through the meticulous separation of objects from their backgrounds, a process executed by selecting the region of interest and leveraging the potent GrabCut algorithm in the OpenCV and Python environments.

To attain precise object dimensions, image pixel calibration was meticulously carried out with reference objects. Furthermore, sophisticated computer vision algorithms were applied for in-depth shape analysis.

For automating the 3D solid model generation within commercial CAD software, a macro was developed. The macro efficiently plotted and connected coordinate points, resulting in the creation of high-quality closed curves.

The overall time required for the entire process ranged from 5 to 15 min. This time frame encapsulated not only the set-up and warm-up duration for the camera and Raspberry PI but also accounted for variations arising from the complexity of the CAD shapes involved. The intricate nature of certain shapes influenced the number of contour points generated and subsequently impacted the overall processing time.

This research exhibits significant potential applications across diverse domains, including 3D design, shape analysis, and 3D printing, due to its ability to automate 3D model generation within CAD packages. Notably, this approach mitigates the demand for extensive CAD proficiency and streamlines the laborious, time-intensive process of measuring multiple object dimensions. Furthermore, it offers valuable implications for quality assurance in 3D printing, where uneven shrinkage can significantly impact the final product.

Additionally, the incorporation of a robotic hand in our methodology provides a means to precisely rotate the object held by the hand and capture finer details of the object. This innovation opens new avenues for object analysis and modeling that hold promise

for addressing persistent challenges in various fields, including quality assurance in 3D printing and advanced object analysis.

## 8. Limitations and Future Works

The primary objective of this study was the generation of 3D solid models from images, laying the foundation for 3D printing applications. While the investigation encompassed objects of both simple and intricate shapes, certain limitations become apparent in the pursuit of our research goals.

One notable limitation lies in the ability to generate 3D models of hollow objects, such as engine blocks or tubes, due to the inherent lack of internal contour information in regular camera images. A potential solution to this limitation could involve the utilization of X-ray imaging, which offers the capacity to capture internal details and thus expand the scope of the presented methodology.

Additionally, the subjects selected for this study primarily consisted of 3D objects with uniform thickness. Consequently, the information extracted from the two images proved adequate for our purposes. However, the prospects for future research hold promise in the examination of more complex objects characterized by irregular shapes with varying depths, blind holes, and non-uniform thickness. This avenue of exploration presents a compelling direction for advancing the capabilities and applicability of the proposed research methodology.

**Author Contributions:** B.B.: Conceptualization, Methodology, Computer Programming, Validation, Formal analysis, Investigation, Writing—original draft, reviewing and editing, and Visualization. P.M.: Reviewing original draft, conceptualization, and methodology. All authors have read and agreed to the published version of the manuscript.

**Funding:** This research received no external funding.

**Data Availability Statement:** The datasets used and/or analyzed during the current study are available from the corresponding author upon reasonable request.

**Acknowledgments:** Special acknowledgment to MinKyo Lee for his assistance in designing the experiment set-up.

**Conflicts of Interest:** Prakash Manandhar is the founder and CEO of Simulate Anything, Inc., Hopkinton, MA 01748, USA. Simulate Anything, Inc. is a venture-backed startup that works on artificial intelligence for engineering design, including generating 3d models from text descriptions of 3d models. The research topic in this publication relates to the reconstruction of 3d models. Prakash Manandhar attests that the results and methodologies were presented based on scientific merit and are not biased by his involvement in Simulate Anything, Inc.

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
