# Peer review of "Integrating Computer Vision and CAD for Precise Dimension Extraction and 3D Solid Model Regeneration for Enhanced Quality Assurance"

_machines, doi:10.3390/machines11121083_

Round 1

Reviewer 1 Report

Comments and Suggestions for Authors

Please see the file attached.

Comments on the Quality of English Language

Minor English grammar and spelling is required.

Author Response

Thank you for your thoughtful and constructive feedback on our manuscript. We appreciate the time and effort you dedicated to reviewing our work. A separate pdf file has been uploaded with a detailed response to your comments. 

Thank you!

Reviewer 2 Report

Comments and Suggestions for Authors

While the article introduces an innovative approach to integrating computer vision and CAD for precise dimension extraction and 3D solid model regeneration, several criticisms can be raised:

1. The conversion of 3D scans into STL files and subsequent 3D printing is a task achievable with existing technology. The author should provide additional academic literature to underscore the novelty of the research and identify areas where the current literature lacks coverage, aspects uniquely addressed by this article.

2. The paper lacks in-depth details on specific computer vision algorithms used, such as GrabCut, Canny edge detector, and morphological operations. A more comprehensive explanation of these algorithms and their parameter choices would benefit readers interested in replicating or extending the study.

3. On page 6, line 191, please correct "Figure 7" to "Figure 5."

4. The author argues that Figure 6 depicts a complex cross-section, but reviewers perceive it as a simple geometric shape. Can the method proposed in this article handle more intricate shapes, such as those with varying depths or shapes with blind holes? Based on the current research framework, it appears that the robotic hand and cameras set up in this study may not accurately capture the dimensions of blind holes.

5. On page 9, the terms "real" and "virtual” measurements in the formulas need to be explained in detail. From Figure 9, what is the error between the virtual measurement obtained from the camera-captured virtual image and the actual object? After the virtual image undergoes processing in a CAD program to become a 3D image, should the tolerance between the physical printout and the 3D image be considered a precision issue with the 3D printer? Does the error obtained at Line 263 imply that this method may not be suitable for engineering drawings with higher precision?

6. In Line 277, "requiring minor geometry adjustments," does this mean manual adjustments are needed? How severe are typical cases of hole breakage, and how should situations with varying depths be addressed?

Author Response

Thank you for your thoughtful and constructive feedback on our manuscript. We appreciate the time and effort you dedicated to reviewing our work. A separate pdf file with a detailed response to the reviewer's comment has been uploaded.
Thank you so much!

Reviewer 3 Report

Comments and Suggestions for Authors

The example is for the case of simple cylindrical shape (with same section all through horizontal plane), but what is happening in case of a real 3D object ?

Author Response

Thank you for your thoughtful and constructive feedback on our manuscript. We appreciate the time and effort you dedicated to reviewing our work. A separate pdf file with a detailed response to the reviewer's comment has been uploaded separately.

Thank you so much!

Round 2

Reviewer 1 Report

Comments and Suggestions for Authors

the manuscript has been properly revised according to the reviewers' comments. 

Author Response

The authors would like to thank the reviewer for his positive comment. The reviewer has mentioned that "the manuscript has been properly revised according to the reviewers' comments" hence no additional revision is needed.

Reviewer 2 Report

Comments and Suggestions for Authors

The reviewer checked the manuscript of machines-2726470-v2 carefully. It is apparent that some contents from the manuscript have been improved according to the reviewer’s suggestions. The second revised comments are recorded as follows:

1. The conversion of 3D scans into STL files and subsequent 3D printing is a task achievable with existing technology. The author should provide additional academic literature to underscore the novelty of the research and identify areas where the current literature lacks coverage, aspects uniquely addressed by this article. 

Reviewer’s 2nd comment: The additional academic literatures have been incorporated within sections I and II to provide a comprehensive overview of the most up-to-date contributions in the studied field.

2. The paper lacks in-depth details on specific computer vision algorithms used, such as GrabCut, Canny edge detector, and morphological operations. A more comprehensive explanation of these algorithms and their parameter choices would benefit readers interested in replicating or extending the study.

Reviewer’s 2nd comment: Some further explanations have been added to assist the readers in understanding of the chosen methodology.

3. On page 6, line 191, please correct "Figure 7" to "Figure 5."

Reviewer’s 2nd comment: It has been corrected via the advice of the reviewer.

4. The author argues that Figure 6 depicts a complex cross-section, but reviewers perceive it as a simple geometric shape. Can the method proposed in this article handle more intricate shapes, such as those with varying depths or shapes with blind holes? Based on the current research framework, it appears that the robotic hand and cameras set up in this study may not accurately capture the dimensions of blind holes.

Reviewer’s 2nd comment: Unable to evaluate.

5. On page 9, the terms "real" and "virtual” measurements in the formulas need to be explained in detail. From Figure 9, what is the error between the virtual measurement obtained from the camera-captured virtual image and the actual object? After the virtual image undergoes processing in a CAD program to become a 3D image, should the tolerance between the physical printout and the 3D image be considered a precision issue with the 3D printer? Does the error obtained at Line 263 imply that this method may not be suitable for engineering drawings with higher precision?

Reviewer’s 2nd comment: No comments. The authors responded that the error at Line 263 indicates a limitation in 3D printing precision. They are actively exploring opportunities to enhance precision and are considering avenues for future research to address higher precision requirements.

6. In Line 277, "requiring minor geometry adjustments," does this mean manual adjustments are needed? How severe are typical cases of hole breakage, and how should situations with varying depths be addressed?

Reviewer’s 2nd comment: It remains uncertain whether the suggested comment can be fully addressed. The authors have stated in their response “In our current research framework, we have acknowledged these limitations and have outlined them in the “Limitations and Future Works” section of the manuscript.”

Author Response

The reviewer checked the manuscript of machines-2726470-v2 carefully. It is apparent that some contents from the manuscript have been improved according to the reviewer’s suggestions. The second revised comments are recorded as follows:

1. The conversion of 3D scans into STL files and subsequent 3D printing is a task achievable with existing technology. The author should provide additional academic literature to underscore the novelty of the research and identify areas where the current literature lacks coverage, aspects uniquely addressed by this article.

Reviewer’s 2nd comment: The additional academic literatures have been incorporated within sections I and II to provide a comprehensive overview of the most up-to-date contributions in the studied field.

The authors appreciate your positive acknowledgement of the efforts made to incorporate additional academic literature within sections I and II. The authors are pleased to note that our efforts to enhance the literature review have been recognized.

2. The paper lacks in-depth details on specific computer vision algorithms used, such as GrabCut, Canny edge detector, and morphological operations. A more comprehensive explanation of these algorithms and their parameter choices would benefit readers interested in replicating or extending the study.

Reviewer’s 2nd comment: Some further explanations have been added to assist the readers in understanding of the chosen methodology.

It is encouraging to learn that the additional explanations provided for the chosen methodology have contributed to a better understanding for the readers.

3. On page 6, line 191, please correct "Figure 7" to "Figure 5."

Reviewer’s 2nd comment: It has been corrected via the advice of the reviewer.

The reviewer seems happy with the revised manuscript which addressed the suggestion from the reviewer.

4. The author argues that Figure 6 depicts a complex cross-section, but reviewers perceive it as a simple geometric shape. Can the method proposed in this article handle more intricate shapes, such as those with varying depths or shapes with blind holes? Based on the current research framework, it appears that the robotic hand and cameras set up in this study may not accurately capture the dimensions of blind holes.

Reviewer’s 2nd comment: Unable to evaluate.

The authors appreciate your diligence in assessing our work during the second revision.

In response to your inquiry about Figure 6 and the capability of our proposed method, the authors want to reiterate our gratitude for your insightful observations. As stated in our initial response and reiterated in the revised manuscript, the current research is indeed limited in handling more intricate shapes such as varying depths and objects with blind holes, we have explicitly outlined this limitations in Section 8, under “Limitations and Future works.”

The revised sentences in the manuscript now emphasize the potential for future research to explore complex objects with irregular shapes, non-uniform thickness, and blind holes. We hope this modification aligns with your expectations and provides clarity on the scope of our current study.

5. On page 9, the terms "real" and "virtual” measurements in the formulas need to be explained in detail. From Figure 9, what is the error between the virtual measurement obtained from the camera-captured virtual image and the actual object? After the virtual image undergoes processing in a CAD program to become a 3D image, should the tolerance between the physical printout and the 3D image be considered a precision issue with the 3D printer? Does the error obtained at Line 263 imply that this method may not be suitable for engineering drawings with higher precision?

Reviewer’s 2nd comment: No comments. The authors responded that the error at Line 263 indicates a limitation in 3D printing precision. They are actively exploring opportunities to enhance precision and are considering avenues for future research to address higher precision requirements.

The reviewer seems convinced with the authors response to his/her comment no.5.

6. In Line 277, "requiring minor geometry adjustments," does this mean manual adjustments are needed? How severe are typical cases of hole breakage, and how should situations with varying depths be addressed?

Reviewer’s 2nd comment: It remains uncertain whether the suggested comment can be fully addressed. The authors have stated in their response “In our current research framework, we have acknowledged these limitations and have outlined them in the “Limitations and Future Works” section of the manuscript.”

The authors would like to express our sincere appreciation for your insightful comments on our manuscript. Your feedback is invaluable, and we are grateful for the time and effort you dedicated to reviewing our work.

Regarding your query on Line 277 regarding “requiring minor geometry adjustments,” we acknowledge the need for clarification. The phrase indicates that, in certain instances, the robotic hand may face challenges in achieving precise object positioning due to limitations in rotational angle adjustment steps. In such cases, we agree that some manual adjustments may be necessary to enhance the accuracy of the regenerated 3D model.

In response to your inquiry about the severity of hole breakage and handling varying depths, we appreciate your keen observation. However, we would like to note that our current research focus has not delved into objects with holes or varying depths. As outlined in the “Limitations and Future Works” section of the manuscript, we have explicitly mentioned these limitations. While we recognize the significance of your suggestion, it falls outside the scope of the present work. Rest assured, we acknowledge the importance of these aspects, and the authors intend to explore these avenues in future research endeavors, as per your valuable recommendations.

Once again, we extend our gratitude for your thoughtful comments, and we hope that our responses adequately address your concerns. Should you have any further inquiries or suggestions, we would be more than willing to discuss and incorporate them into our work.